# The impact of environmental enrichment on energy metabolism in ovariectomized mice

Chaoran Ju[1], Ayano Ogura[1,2], Yoshikazu Hayashi[3,4,5], Yuko Kawabata[6], Fulvio D'Acquisto[7,8], Tomoyo Kawakubo-Yasukochi[3]*, Eijiro Jimi[1,3]*

1 Laboratory of Molecular and Cellular Biochemistry, Division of Oral Biological Sciences, Faculty of Dental Science, Kyushu University, Fukuoka, Japan, 2 Division of Oral Biological Sciences, Department of Cell Biology, Aging Science, and Pharmacology, Faculty of Dental Science, Kyushu University, Fukuoka, Japan, 3 Oral Health/Brain Health/Total Health Research Center, Faculty of Dental Science, Kyushu University, Fukuoka, Japan, 4 Division of Functional Structure, Department of Morphological Biology, Fukuoka Dental College, Fukuoka, Japan, 5 Oral Medicine Research Center, Fukuoka Dental College, Fukuoka, Japan, 6 Section of Oral Neuroscience, Division of Oral Biological Sciences, Faculty of Dental Science, Kyushu University, Fukuoka, Japan, 7 School of Life and Health Science, University of Roehampton, London, United Kingdom, 8 William Harvey Research Institute, School of Medicine and Dentistry, Queen Mary University of London, London, United Kingdom

* tomoyo@dent.kyushu-u.ac.jp (TK-Y); ejimi@dent.kyushu-u.ac.jp (EJ)

## Abstract

After menopause, a decline in ovarian function leads to various physical and psychological changes, potentially resulting in a range of pathological conditions, including abnormalities in energy metabolism. In recent years, environmental enrichment, which is characterized by positive and comfortable eustress, has been shown to improve various physiological and pathological conditions. This study investigated the effects of environmental factors on energy metabolism in a menopause model using an ovariectomized (OVX) mouse model. Wild-type female mice (8-week-old) were subjected to OVX or a sham operation and maintained under standard condition (SC), enriched environment (EE), or isolated (IS) condition for 4 weeks. OVX led to weight gain and disruption of circadian rhythms, along with changes in various metabolic parameters influenced by differences in housing environments; i.e., EE improved metabolic parameters, but IS deteriorated them. Physical activity and social interaction were factors that determined these differences. Menopause is usually a significant transitional period in a woman's life, and changes in the social environment during this period can contribute to a diverse range of physical and psychological symptoms. Consequently, when implementing interventions to alleviate menopause-related pathological conditions, not only physical symptoms but also the social context should be carefully considered.

## Introduction

It has been suggested that the living environment may affect not only the physiological status [1] but also the progression of diseases [2], although environmental enrichment in humans remains controversial and is difficult to analyze in detail because there is no standardized environment. In laboratory animals, a housing environment that includes social interaction and physical exercise in space use is called environmental enrichment, and it has been reported to be effective in eliciting animals' innate behavior and improving behavior [3],

**Data availability statement:** All relevant data are within the paper and its Supporting Information files.

**Funding:** the Japan Society for the Promotion of Science (JSPS) Grants-in-Aid for Scientific Research (KAKENHI) (Grant No. 20KK0213 to EJ).

**Competing interests:** The authors have declared that no competing interests exist.

**Abbreviations:** OVX, ovariectomy; SC, standard condition; EE, enriched environment; IS, isolated condition; gWAT, gonadal white adipose tissue; ELISA, enzyme-linked immunosorbent assay; H&E, Mayer's hematoxylin and eosin Y; GTT, glucose tolerance test; PTT, pyruvate tolerance test; HOMA-IR, Homeostatic Model Assessment for Insulin Resistance; AUC, area under the curve.

neurological functions [3], energy metabolism [4], and cancer progression [4,5]. Therefore, the living environment has been recognized as a research subject [3].

Along with the changing social structure, households are becoming smaller and aging, and the number of elderly people living alone is increasing in Western countries and Japan [6]. Living alone increases the risk of social isolation and shortens healthy life expectancy [7]. Therefore, there is an urgent need to present scientific evidence that living environment is related to the onset or progression of the disease to propose strategies for effective interventions and disease prevention. In particular, postmenopausal women are known to have an increased risk of developing various lifestyle-associated diseases originally regulated by estradiol, such as osteoporosis, obesity, and heart disease [8]. However, no detailed studies have examined the influence of environmental factors on various diseases in postmenopausal women. Proving the hypothesis that the living environment has a significant impact on the risk of developing metabolic diseases after menopause will highlight the importance of environmental enrichment from a preventive medicine perspective. Therefore, in this study, we investigated whether environmental factors, such as environmental enrichment or isolation, affect energy metabolism abnormalities after menopause.

## Material and methods

### Mice

All animal experiments were approved by the Animal Care Committee of Kyushu University (approval numbers A21-358 and A23-168).

### Experimental animals and environmental conditions

C57BL/6J female mice, obtained from CLEA Japan (Tokyo, Japan), were raised in specific-pathogen-free facility under controlled environmental conditions (a 12-hours light/dark cycle, temperature of 20–26 °C, and humidity of 40–70%). The mice at 8-week-old were randomized by body weight and assigned to either the enriched environment (EE), the standard condition (SC), or the isolated condition (IS). The EE mice (5 mice/cage) were housed in a cage (267[W] × 426[D] × 150[H] mm) consisted of a running wheel and shelter (Mouse lgloo and Fast Tracs, Bio-Serv, Flemington, NJ, USA) and a tunnel (Shepherd Tube, EP Trading, Tokyo, Japan), in which a bedding mixture of 2:1 soft bedding (Alpha-dri-Certified, EP Trading) and woody tips (Tokojiki, Douourika, Sapporo, Japan) was used. The SC (5 mice/cage) and IS mice (1 mouse/cage) were maintained in standard cages (168[W] × 299[D] × 133[H] mm) with no additional objects, in which only woody tips (Tokojiki, Douourika) were used. All bedding in the cages was replaced weekly. The mice were either sham-operated (sham) or ovariectomized (OVX) at 9 weeks of age under anesthesia using intraperitoneal injections of 0.75 mg/kg medetomidine, midazolam (4 mg/kg), and saline (5 mg/kg) [9].

### Histological analysis

White adipose tissue was fixed with 4% paraformaldehyde and embedded in paraffin. For histological examination, Mayer's hematoxylin and eosin Y staining was applied to each 5 μm-thick section and observed using a microscope (BZ-X800, KEYENCE, Osaka, Japan). The adipocyte area was measured using the BZ-X800 Analyzer software (KEYENCE) [10].

### Glucose (GTT) and pyruvate (PTT) tolerance tests

At 12 weeks of age, the mice were subjected to an intraperitoneal injection of glucose (2 mg/kg body weight) or pyruvate solution (Nacalai Tesque, Kyoto, Japan, 1.5 g/kg body weight)

after 20 hours of fasting. Glucose levels were assessed using a handheld glucometer (Freestyle Freedom Lite, Abbott Park, IL, USA) at 0, 15, 30, 60, and 90 minutes after injection.

## Homeostasis model assessment of insulin resistance (HOMA-IR)

The HOMA-IR index was calculated using Equation (1) [11]:

$$HOMA-IR = \text{fasting serum glucose } (mg/dl) \times \text{fasting serum insulin } (ng/ml) \times 26/405 \tag{1}$$

Serum glucose and insulin levels after fasting for 20 hours were measured using a D-glucose assay kit (GOPOD Format; Megazyme, Wicklow, Ireland) and an LBIS™ Mouse Insulin ELISA Kit (U-type) (Fujifilm Wako, Osaka, Japan), respectively.

## Serum corticosterone measurement

Serum corticosterone levels were measured using the DetectX Corticosterone ELISA Kit (Arbor Assays, Ann Arbor, MI, USA) according to the manufacturer's instructions.

## Serum biochemical analysis

All serum biological analysis was performed by Oriental Yeast (Tokyo, Japan).

## Measurement of mouse momentum and body temperature

The amount of activity and subcutaneous temperature in the mice were monitored using an implantable device (nanotag, Kissei Comtec, Matsumoto, Japan) and analyzed using the nanotag Viewer program (Kissei Comtec) [12].

## Measurement of daily food consumption

Daily feed intake was measured using a powder feeder and multifeeder (MF3S for an IS cage and MF4S for SC and EE cages) (SHINFACTORY, Fukuoka, Japan).

## Statistical analysis

ANOVA followed by Tukey-Kramer test was performed using GraphPad Prism 10 software (version 10.4.1; GraphPad Software, San Diego, CA, USA). Each parametric test was followed by a multiple comparison test after applying the D'Agostino & Pearson test to determine and validate the presence of a normal distribution. All experiments were performed more than three times, and all quantitative data are presented as mean ± standard error of the mean (SE). P values < 0.05 were considered statistically significant.

# Results

## Effects of body weight change and housing environment in sham and OVX mice

To evaluate the effects of different environments on sham and OVX menopausal model mice, mice were randomly assigned to SC, EE, and IS environments to allow them to acclimate to the environment one week before the operation (sham or OVX) (Fig 1A, 1B). At the time of the sham/OVX operation (week 1), no difference in body weight was observed among the groups (Fig 1C). Sham and OVX mice raised for an additional 3 weeks (week 4) showed no marked differences in body weight between the SC, EE, and IS environments (Fig 1D). OVX-induced weight gain was observed in each housing environment (Fig 1D), which

was consistent with a previous report that OVX mice had higher body weights than sham-operated mice [13]. The difference in body weight gain between weeks 1 and 4 also indicated that OVX led to weight gain regardless of the environment (Fig 1E).

## Effects of housing environment on feed intake, energy expenditure, and body temperature in sham-operated and OVX mice

Body weight gain is closely related to an energy imbalance between energy intake and expenditure [14]. Therefore, we investigated whether there were differences in feed intake, physical activity, and body temperature between these mice in the different environments. Daily feed intake was monitored during week 3. Regardless of whether they were in the sham or OVX

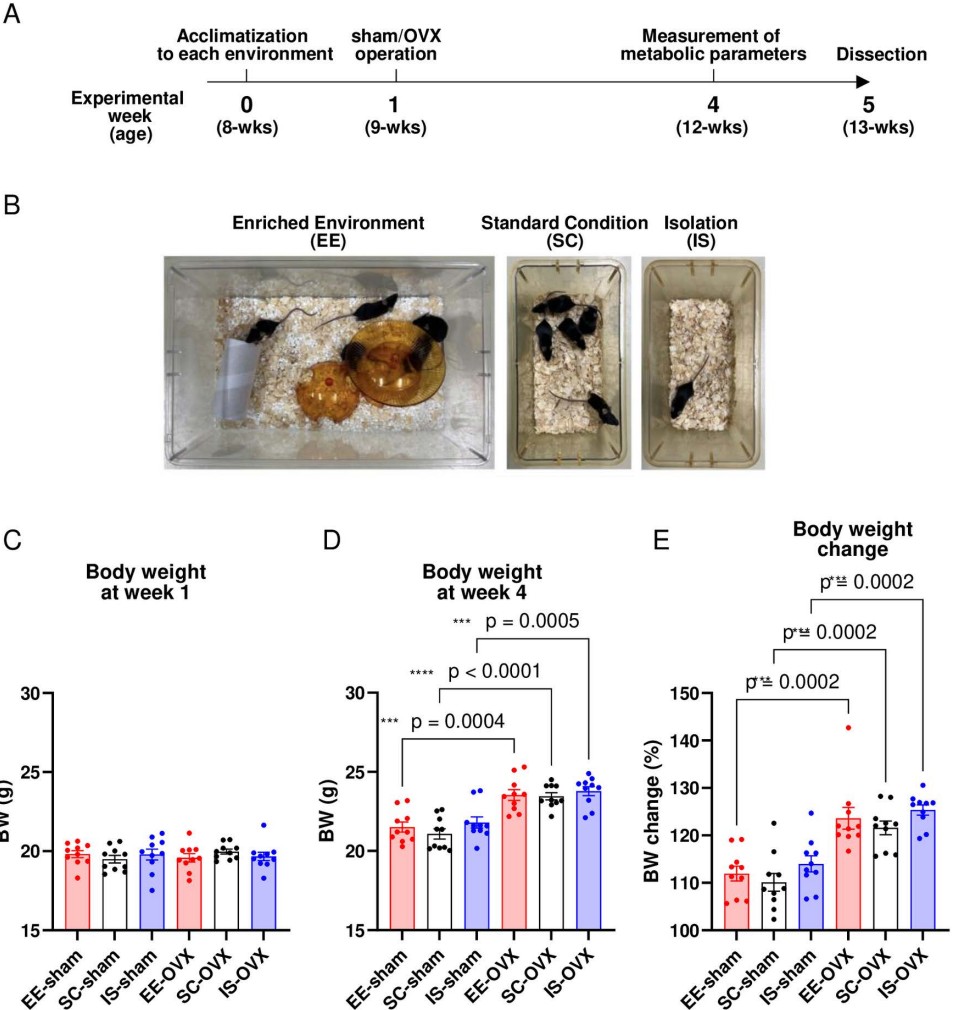

**Fig 1. Housing environments and body weight changes.** (A) Experimental schedule. (B) Images of three different housing environments, (Left) EE, (Middle) SC, and (Right) IS environments. (C, D) Body weight at week 1 (C) and week 4 (D). (E) Body weight changes in (C) and (D). In all groups, n = 10. Experiments were repeated at least three times with similar results. All values are shown as the means ± SE. One-way ANOVA and Tukey's multiple comparisons test were used; ***$P < 0.001$, ****$P < 0.0001$. SE, standard error; ANOVA, analysis of variance; OVX, ovariectomized; SC, standard conditions; EE, enriched environment; IS, isolated conditions.

group, the feed intake of SC mice was substantially higher than that of EE mice (Fig 2A). In addition, OVX led to a higher increase in feed intake in IS than in EE mice (Fig 2A).

In the sham group, the physical activity level of EE mice in week 3 was substantially higher than that in the SC and IS groups; however, that in the EE-sham mice was markedly lower after OVX (Fig 2B, 2C), which is consistent with the fact that the decrease in estradiol reduces physical activity [15]. Estradiol regulates circadian rhythm [15]. The circadian rhythm of physical activity observed in sham mice, which indicates decreased activity during the light period and increased activity during the dark period, was not clearly observed in the OVX group (Fig 2B, 2C).

The circadian rhythm of body temperature, which is a vital metabolic sign, was also disrupted in the OVX group (Fig 2D). In addition, a marked decrease in body temperature was observed in the IS-sham group compared to that in the EE-sham group, especially during the light period (Fig 2E–2G). No changes in body temperature were observed in the OVX groups in the housing environments (Fig 2D–2G).

## Effects of housing environment on adipose tissue phenotype in sham-operated and OVX mice

To investigate the cause of weight gain in OVX mice (Fig 1E), gonadal white adipose tissue (gWAT) was analyzed 4 weeks after environmental habituation (weeks 5, Fig 1A). A substantial increase in gWAT mass in OVX mice compared to that in sham mice in each corresponding environment was observed (Fig 3A), suggesting that OVX-related body weight change was closely related to gWAT weight change. Furthermore, gWAT weight was substantially lower in EE mice than in the sham and OVX groups (Fig 3A), whereas no change in body weight was found in the latter groups (Fig 1C–1E). This tendency was negatively correlated with physical activity (Fig 2B, 2C), suggesting that exercise might alter the quantity and quality of gWAT. The histological analysis indicated that the adipocytes of EE mice were substantially smaller than those of SC mice in the sham and OVX groups (Fig 3B, 3C), and there was a positive correlation between gWAT weight (Fig 3A) and adipocyte size (Fig 3B, 3C). These results suggest that gWAT expansion was mainly caused by adipocyte hypertrophy.

## Effects of housing environment on glucose metabolic parameters in sham and OVX mice

Since white adipose tissues play an essential role in glucose homeostasis [16,17], glucose metabolism disorders are commonly found during menopause [18]. Therefore, the effects of the environment on glucose metabolic parameters in sham and OVX mice were investigated.

In glucose tolerance tests (GTT), IS-sham mice showed decreased glucose tolerance compared to mice in EE and SC, although there was no statistical difference between sham and OVX mice in the IS environment (Fig 4A, 4B). The most characteristic trend was an increase in blood glucose levels 15 minutes after glucose injection and fasting glucose levels under IS conditions in both sham and OVX mice (Fig 4C, 4D), although there was no difference in leisure blood glucose levels among the groups (Fig 4E). Severe postprandial glucose spike and high fasting glucose levels adversely affect long-term glycemic control and increase the risk of cardiovascular events, and which can lead to more damage than a steady state of high glucose [19–21].

Menopausal mouse models exhibit insulin resistance and hyperinsulinemia [22]. In the current study, the leisure serum insulin concentration in OVX mice was substantially higher than that in sham mice in SC and IS, and this tendency was improved in EE (Fig 1F). In contrast, no marked difference was observed in fasting serum insulin levels between sham and

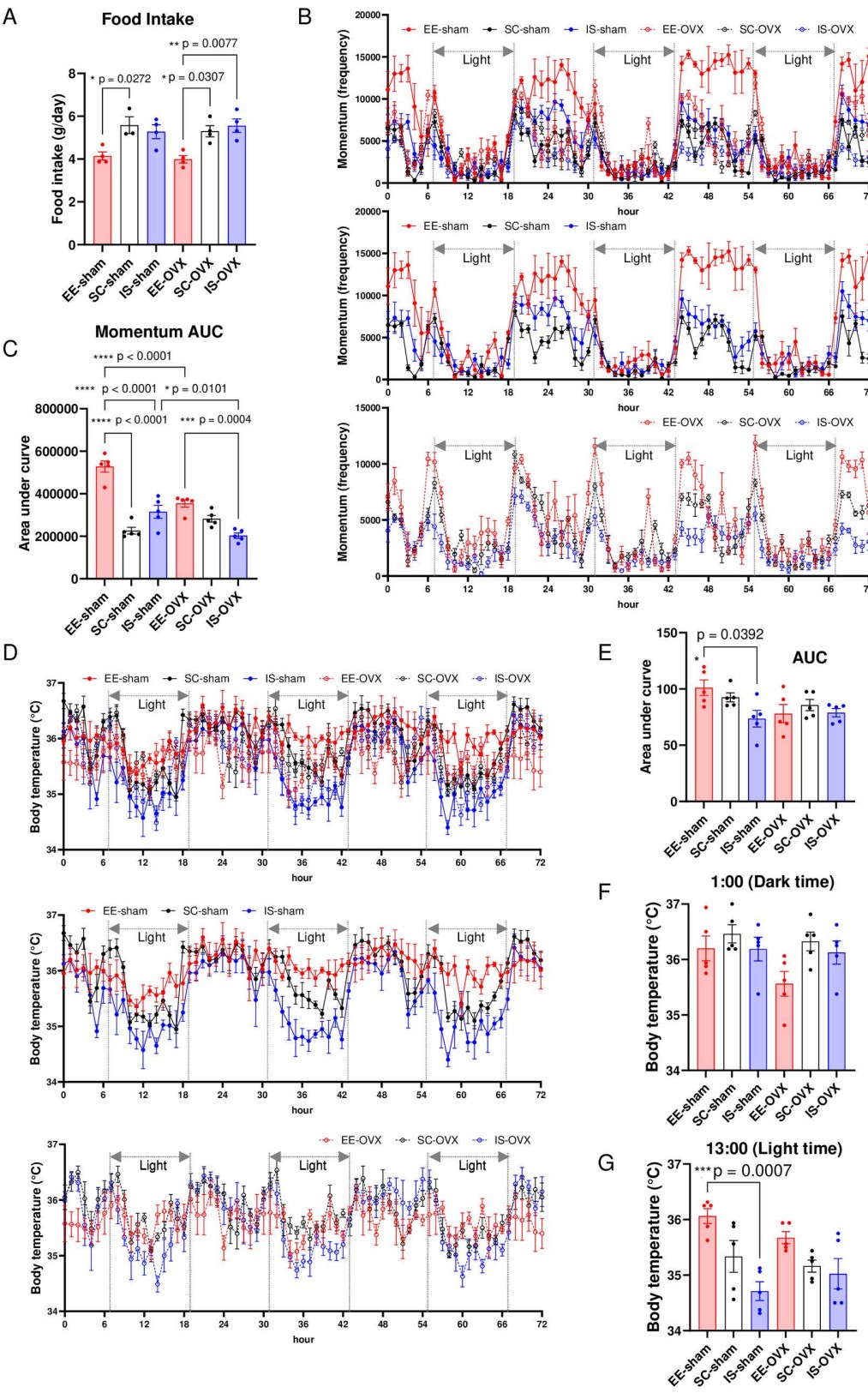

**Fig 2. Measurements of feed intake, physical activity, and body temperature.** (A) Daily feed intake per mouse. (B) Momentum per mouse. (C) Area under the curve for (B). (D) Body temperature during week 3. (E) Area under the curve for (D). (F,G) Body temperature at 01:00 (F) and 13:00 (G). Red: EE, Black: SC, Blue: IS. Experiments were

repeated at least three times with similar results. All values are shown as the means ± SE. One-way ANOVA and Tukey's multiple comparisons test were used; *P < 0.05, **P < 0.01, ***P < 0.001, ****P < 0.0001. SE, standard error; ANOVA, analysis of variance; OVX, ovariectomized; SC, standard conditions; EE, enriched environment; IS, isolated conditions.

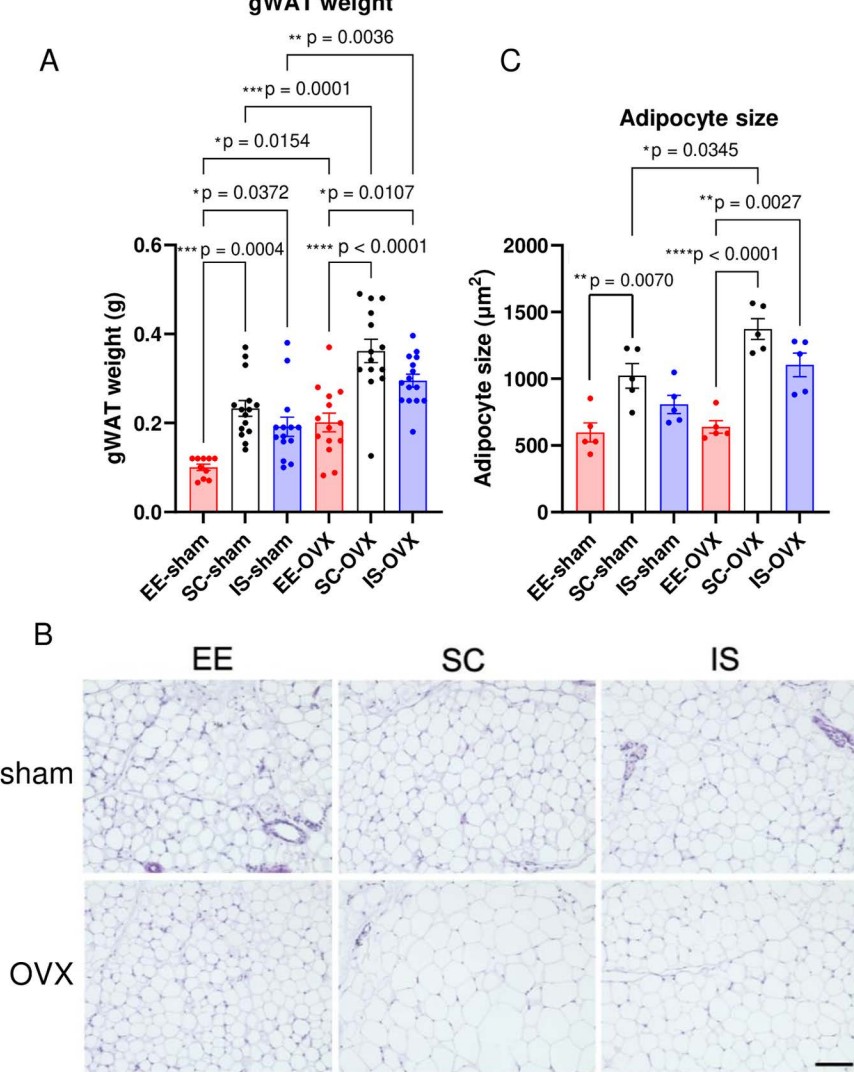

**Fig 3. Measurements of white adipose tissue weight and adipocyte size.** (A) Gonadal white adipose tissue (gWAT) weight at week 5 (n = 14-15). (B) Representative images of H&E staining in gWAT. Scale bars: 100 μm. (C) Quantification of the adipocyte area (area per adipocyte, μm2) performed using a microscope at 20× magnification in four random fields per section (n = 5). The experiments were repeated at least three times, and similar results were obtained. All values are shown as the means ± SE. One-way ANOVA and Tukey's multiple comparison tests were used; *P < 0.05, **P < 0.01, *** P < 0.001, **** P < 0.0001. SE, standard error; ANOVA, analysis of variance; OVX, ovariectomized; South Carolina, standard conditions; EE, enriched environment; IS, isolated conditions.

OVX mice (Fig 4G), as previously reported [23], but it was decreased in EE compared to IS in the OVX groups (Fig 4G).

HOMA-IR, an index of insulin resistance calculated from fasting serum glucose and fasting insulin concentrations, was substantially elevated in the IS compared to EE in the OVX mice (Fig 4H).

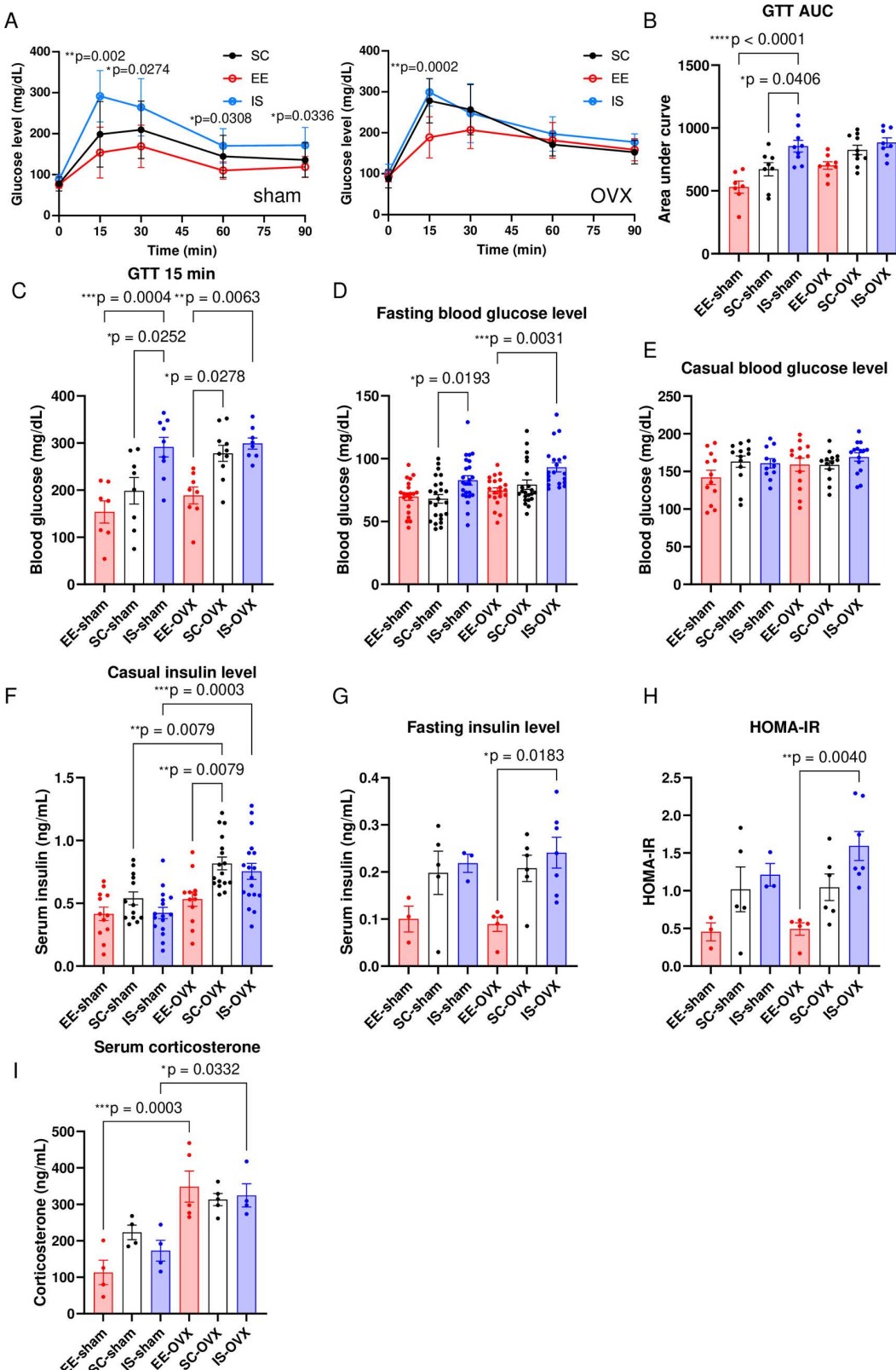

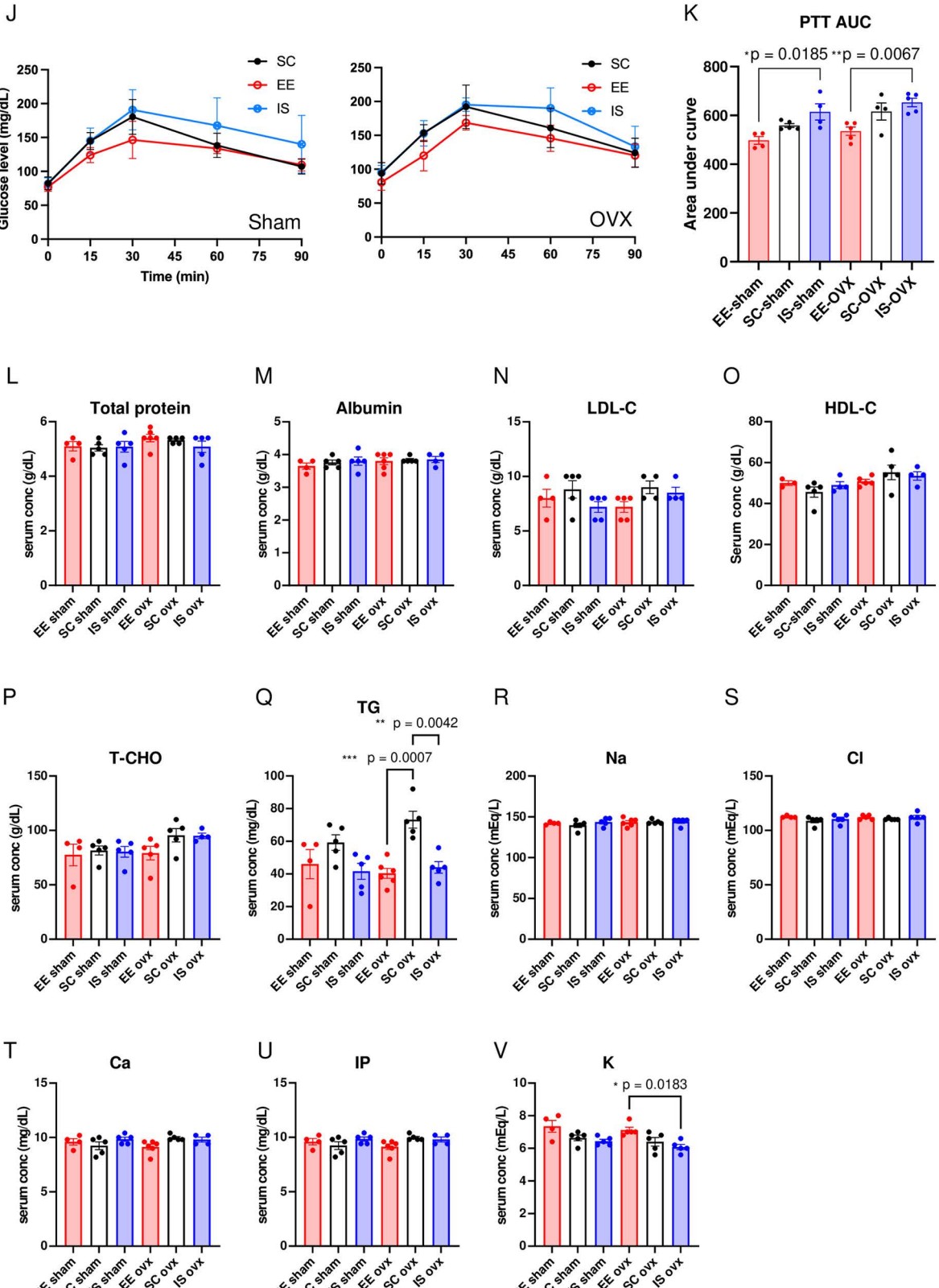

**Fig 4. Measurement of metabolic parameters and serological analysis.** (A) GTT in sham (left) and OVX (right) mice (n = 5). (B) Area under the curve for (A). (C) Blood glucose level 15 minutes after glucose injection during GTT. (D) Fasting blood glucose level. (E) Leisure blood glucose level. (F) Leisure serum insulin level. (G) Fasting serum insulin level. (H) HOMA-IR value. (I) Serum corticosterone

level. (J) PTT in sham (left) and OVX (right) mice (n = 5). (K) Area under the curve for (J). (L–V) Serological tests for total protein (L), Albumin (M), LDL-cholesterol (N), HDL-cholesterol (O), T.-cholesterol (P), triglyceride (Q), sodium (R), chloride (S), calcium (T), inorganic phosphorus (U), and potassium (V). Experiments were repeated at least three times with similar results. All values are shown as the means ± SE. One-way ANOVA and Tukey's multiple comparisons test were used, *$P < 0.05$, **$P < 0.01$, *** $P < 0.001$. SE, standard error; ANOVA, analysis of variance; OVX, ovariectomized; South Carolina, standard conditions; EE, enriched environment; IS, isolated conditions; HOMA-IR, homeostasis model assessment of insulin resistance; glucose (GTT) and pyruvate (PTT) tolerance tests; L/HDL, low/high density lipoprotein.

A strong relationship exists between glucose regulation and glucocorticoids (24), which are the main effectors of stress response [24,25] A study reported that OVX increased serum glucocorticoid content compared to sham-operated mice [26], but there is a report of no change in these contents [27]. In the current study, we observed an OVX-induced increase in corticosterone levels in the EE and IS groups, although there was no change in serum corticosterone content due to environmental differences (Fig 4I). Serum corticosterone contents in mice other than EE-sham mice increased over the reference range, suggesting that these mice might be in a stressful environment [28].

Gluconeogenesis is partially regulated by glucocorticoids [29–31]; therefore, gluconeogenesis was studied in in the mice. Glycemic excursion in response to pyruvate during pyruvate tolerance tests (PTT) was suppressed in the EE compared to the IS environment in the sham and OVX groups (Fig 4L, 4M).

Serological test results showed no statistical differences between the groups in total protein, albumin, LDL-cholesterol (LDL-C), HDL-cholesterol (HDL-C), total cholesterol (T-CHO), Sodium (Na), Chloride (Cl), Calcium (Ca), and Inorganic phosphorus (IP), whereas triglyceride (TG) was substantially increased in SC compared EE and IS groups in OVX mice (Fig 4N–4W). In addition, the serum potassium levels were markedly lower in IS-OVX than in EE-OVX mice (Fig 4U). Given that insulin is a regulator of potassium intracellular uptake [32], high serum insulin concentrations (Fig 4G, 4F) can be correlated with decreased serum potassium levels in IS-OVX mice (Fig 4U).

## Discussion

In the current study, we proved that OVX induced weight gain and circadian rhythm disruption, and that metabolic parameters changed depending on the housing environment; EE was better and IS was worse. Therefore, glucose intolerance and insulin resistance were the most affected in IS-OVX mice.

Exercise is an important environmental factor. We suggest that exercise in EE induces increased muscle mass and fat burning, and improves metabolic parameters. Although the space allocation per mouse was greater in the IS than in the SC environment, there was no difference in physical activity levels between the SC and IS. The reason for the deterioration in energy metabolism in IS-OVX mice over SC-OVX mice was likely due to decreased social enrichment. Although housing densities are also known to be important for animal welfare [33], social isolation had a major impact on metabolic abnormalities in IS-OVX mice in this study.

No definitive correlations were found between feed intake, exercise, and body weight. It is generally known that overeating and a lack of exercise are not direct causes of body weight gain [34], and that feed intake, physical activity, and body weight changes are influenced by multiple factors, including stress [34], such that further studies on endocrine factors are required. The changes in body weight and gWAT were not always correlated in all groups. Changes in fat mass also affect changes in lean muscle mass. Exercise promotes an increase in muscle mass [35] and fat burning [36]; changes in both muscle and fat mass lead to changes in

body weight. Therefore, further analyses of body composition may be required to determine whether changes in body weight are dependent on fat or lean mass in all mice.

Furthermore, changes in body temperature owing to stress have long been reported [37]. In the current study, body temperature was substantially lower in IS-sham than in EE-sham mice (Fig 2E–2G), but no differences in corticosterone levels were observed between the groups (Fig 4I). The temperature of the experimental room was set to 20–26 °C, in accordance with international guidelines [38]. However, this temperature environment can be cold stress for rodents, in which the body temperature can easily drop if they are constantly active [38]. Therefore, body temperature results can be attributed to the differences between groups and individual housing.

## Conclusion

This study demonstrated that in a housing environment in which exercise and social contact are eliminated, energy metabolism abnormalities can occur within a short period of time. OVX disrupts circadian rhythms, resulting in a decrease in physical activity even in an enriching environment; social isolation worsens metabolic abnormalities such as insulin resistance. There are some limitations in this study. For example, we observed the effects of comprehensive environmental enrichment without distinguishing between physical and social enrichment, and all animals were analyzed after invasive surgery, OVX or sham, which might make them to feel some stress. However, even considering the limitations of our experimental design, our results can suggest that an enriched living environment with societal contact and activity are important for promoting health, especially after menopause.

## Supporting information

**S1 Raw Data. Fig 2B & D_4A&J raw data.**
(XLSX)

## Author contributions

**Conceptualization:** Fulvio D'Acquisto, Eijiro Jimi.

**Formal analysis:** Ayano Ogura, Yoshikazu Hayashi, Yuko Kawabata.

**Funding acquisition:** Eijiro Jimi.

**Investigation:** Chaoran Ju, Ayano Ogura, Yoshikazu Hayashi, Yuko Kawabata, Tomoyo Kawakubo-Yasukochi.

**Methodology:** Chaoran Ju, Fulvio D'Acquisto, Eijiro Jimi.

**Supervision:** Tomoyo Kawakubo-Yasukochi.

**Validation:** Ayano Ogura, Yoshikazu Hayashi, Tomoyo Kawakubo-Yasukochi.

**Visualization:** Chaoran Ju, Yoshikazu Hayashi.

**Writing – original draft:** Chaoran Ju, Tomoyo Kawakubo-Yasukochi.

**Writing – review & editing:** Yoshikazu Hayashi, Yuko Kawabata, Fulvio D'Acquisto, Tomoyo Kawakubo-Yasukochi, Eijiro Jimi.

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
