## [Decision Letter · Decision Letter 0]

5 Feb 2025

PONE-D-25-01987The impact of environmental enrichment on energy metabolism in ovariectomized micePLOS ONE

Dear Dr. Jimi,

Thank you for submitting your manuscript to PLOS ONE. After careful consideration, we feel that it has merit but does not fully meet PLOS ONE’s publication criteria as it currently stands. Therefore, we invite you to submit a revised version of the manuscript that addresses the minor points raised by the reviewer.

We look forward to receiving your revised manuscript.

Kind regards,

Andre van Wijnen

Academic Editor

PLOS ONE

Journal Requirements:

2. Thank you for stating the following financial disclosure: the Japan Society for the Promotion of Science (JSPS) Grants-in-Aid for Scientific Research (KAKENHI) (Grant No. 20KK0213 to EJ). 

3. Thank you for stating the following in the Acknowledgments Section of your manuscript: the Japan Society for the Promotion of Science (JSPS) Grants-in-Aid for Scientific Research (KAKENHI) (Grant No. 20KK0213 to EJ). 

Please remove any funding-related text from the manuscript and let us know how you would like to update your Funding Statement. Currently, your Funding Statement reads as follows: the Japan Society for the Promotion of Science (JSPS) Grants-in-Aid for Scientific Research (KAKENHI) (Grant No. 20KK0213 to EJ). 

Reviewers' comments:

Reviewer's Responses to Questions

**Comments to the Author**

1. Is the manuscript technically sound, and do the data support the conclusions?

Reviewer #1: Yes

2. Has the statistical analysis been performed appropriately and rigorously? 

Reviewer #1: No

3. Have the authors made all data underlying the findings in their manuscript fully available?

Reviewer #1: Yes

4. Is the manuscript presented in an intelligible fashion and written in standard English?

Reviewer #1: Yes

5. Review Comments to the Author

Reviewer #1: Dear Authors,

Strictly follow my comments/correction in the attached file. In general:

1. at the end of the introduction section, need of the study should be ideally raised followed by the hypothesis of the study.

2. in statistical analysis, add mathematical model for better understanding of data analysis.

3. in results, add actual p-value of each result

4. in conclusion, add limitations of the study

Thank you!

6. PLOS authors have the option to publish the peer review history of their article (what does this mean? ). If published, this will include your full peer review and any attached files.

**Do you want your identity to be public for this peer review?** For information about this choice, including consent withdrawal, please see our Privacy Policy .

Reviewer #1: No

---

## [Author Response · Author response to Decision Letter 1]

9 Feb 2025

We sincerely appreciate the thorough analyses and constructive suggestions provided by the reviewer, which have been very helpful in guiding us to improve our study further. We hope the editor and the reviewer will concur with us after reading the enclosed point-to-point response that we have addressed all the raised concerns satisfactorily and that the revised manuscript is now suitable for its publication as a Research Article in PLOS ONE.

To Reviewer #1:

Dear Authors,

Strictly follow my comments/correction in the attached file. In general:

1. at the end of the introduction section, need of the study should be ideally raised followed by the hypothesis of the study.

We appreciate the reviewer’s comment. We added a statement of the need for this research and a hypothesis at the end of the “Introduction” section (page2, Line 65-68) in the revised manuscript.

2. in statistical analysis, add mathematical model for better understanding of data analysis.

In agreement with the reviewer’s suggestion, we revised the “Materials and Methods” section (page 4, Line 126-132) in the revised manuscript.

3. in results, add actual p-value of each result

According to the reviewer’s suggestion, we added respective actual p-values to all graphs in which significant differences were obtained. We added p-values in the “Figure Legends” section (lines 156, 190, and 218) in the revised manuscript.

4. in conclusion, add limitations of the study

According to the reviewer’s suggestion, we added limitations of the study in the “Conclusion” section (page 14, Line 325-329) in the revised manuscript.

We also revised the manuscript about the following minor points that the reviewer kindly pointed out.

5. Page 4, Line 91-93, in the original manuscript, which experimental design was followed?

We revised the sentence to “The mice at 8-week-old were randomized by body weight and assigned to either the enriched environment (EE), the standard condition (SC), or the isolated condition (IS)” in the “Materials and Methods” section (page 3, Line 79-80) in the revised manuscript.

6. Page 15, Line 376-379, in the original manuscript, Add reference.

The appropriate reference #10 was added to the “Material & methods” section on page 3, line 98, and to the "References" section on p15, lines 276-379, in the revised manuscript.

---

## [Decision Letter · Decision Letter 1]

17 Feb 2025

The impact of environmental enrichment on energy metabolism in ovariectomized mice

PONE-D-25-01987R1

Dear Dr. Jimi,

We’re pleased to inform you that your manuscript has been judged scientifically suitable for publication and will be formally accepted for publication once it meets all outstanding technical requirements.

Kind regards,

Andre van Wijnen

Academic Editor

PLOS ONE

Additional Editor Comments (optional):

Reviewers' comments:

Reviewer's Responses to Questions

**Comments to the Author**

1. If the authors have adequately addressed your comments raised in a previous round of review and you feel that this manuscript is now acceptable for publication, you may indicate that here to bypass the “Comments to the Author” section, enter your conflict of interest statement in the “Confidential to Editor” section, and submit your "Accept" recommendation.

Reviewer #1: All comments have been addressed

2. Is the manuscript technically sound, and do the data support the conclusions?

Reviewer #1: Yes

3. Has the statistical analysis been performed appropriately and rigorously? 

Reviewer #1: Yes

4. Have the authors made all data underlying the findings in their manuscript fully available?

Reviewer #1: Yes

5. Is the manuscript presented in an intelligible fashion and written in standard English?

Reviewer #1: Yes

6. Review Comments to the Author

Reviewer #1: (No Response)

7. PLOS authors have the option to publish the peer review history of their article (what does this mean? ). If published, this will include your full peer review and any attached files.

**Do you want your identity to be public for this peer review?** For information about this choice, including consent withdrawal, please see our Privacy Policy .

Reviewer #1: No

---

## [Editor Report · Acceptance letter]

PONE-D-25-01987R1

PLOS ONE

Dear Dr. Jimi,

I'm pleased to inform you that your manuscript has been deemed suitable for publication in PLOS ONE. Congratulations! Your manuscript is now being handed over to our production team.

Kind regards,

on behalf of

Dr. Andre van Wijnen

Academic Editor

PLOS ONE